# A recombinase-activated ribozyme to knock down endogenous gene expression in zebrafish

Thomas Juan[1,2,3,4]*, Tonatiuh Molina[1], Lihan Xie[1], Sofia Papadopoulou[1], Bárbara Cardoso[1], Shivam Govind Jha[4], Didier Y.R. Stainier[1,2,3]*

1 Max Planck Institute for Heart and Lung Research, Department of Developmental Genetics, Bad Nauheim, Germany, 2 German Centre for Cardiovascular Research (DZHK), Partner Site Rhine-Main, Bad Nauheim, Germany, 3 Cardio-Pulmonary Institute (CPI), Bad Nauheim, Germany, 4 Department of Immunology, Genetics and Pathology, Uppsala University, Uppsala, Sweden

* thomas.juan@igp.uu.se (TJ); didier.stainier@mpi-bn.mpg.de (DYRS)

## Abstract

Precise regulation of gene expression is essential to understand a wide range of biological processes. Control over gene expression can be achieved using site-directed recombinases and endonucleases whose efficiency is variable and dependent on the genomic context. Here, we develop a self-cleaving ribozyme-based tool to control mRNA levels of endogenous targets in zebrafish. Using an *in vivo* reporter strategy, we first show that inserting the *T3H48* self-cleaving ribozyme in an intron enables rapid pre-mRNA cleavage, with up to 20-fold reduction in expression, and that this ribozyme displays superior activity compared with other ribozymes. We then inserted the *T3H48* ribozyme in the second intron of the *albino* gene using a CRISPR/Cas9 strategy and observed a pigmentation phenotype similar to that in the mutant. Using a base-editing strategy to inactivate the ribozyme, we also show that this phenotype is reversible, illustrating the specificity of the approach. In addition, we generated a Flippase- and Cre-activatable version of the *T3H48* ribozyme, called RiboFlip, to control the mRNA levels of the *albino* gene. RiboFlip activation induced mRNA knockdown and also recapitulated the *albino* mutant phenotype. Furthermore, we show that a Cre- and Dre-controllable Gal4/UAS reporter in the RiboFlip cassette can label knocked-down cells independently of the expression of the target gene. Altogether, we introduce the RiboFlip cassette as a flexible tool to control endogenous gene expression in a vertebrate model and as an alternative to existing conditional knockdown strategies.

## Author summary

Understanding mRNA and protein function requires tools to module gene expression. Current tools in vertebrates include site-directed recombinases and endonucleases, which display variable efficiency depending on the genomic context. Here, we introduce a complementary tool to knock down gene expression in zebrafish, one based on the *T3H48* self-cleaving ribozyme. We first show that the *T3H48* ribozyme can reduce the expression of a reporter transgene, as well as that of an endogenous gene. Using a base-editing

**Data availability statement:** All relevant data are within the manuscript and its Supporting Information files.

**Funding:** This work was supported by a European Molecular Biology Organization Long-Term Fellowship (ALTF 1234-2018) to Thomas Juan, and funds from the Max Planck Society and an award from the European Research Council (ERC) under the European Union's research and innovation programmes (AdG 101021349-TAaGC) to Didier Stainier. The funders had no role in study design, data collection and analysis, decision to publish, or preparation of the manuscript.

**Competing interests:** The authors have declared that no competing interests exist.

strategy to inactivate the ribozyme, we show that this knock down is specific and reversible. We then created a Flippase- and Cre-activatable *T3H48* ribozyme called RiboFlip. We find that the induction of RiboFlip recapitulates the mutant phenotype when inserted in the *albino* gene. Moreover, we show that a Cre- and Dre-controllable Gal4/UAS reporter in the RiboFlip cassette can label knocked-down cells independently of the expression of the target gene. Altogether, these data show that the RiboFlip can serve as a flexible knockdown tool, thereby complementing existing strategies to control gene expression.

## Introduction

Modulating RNA and protein expression using genetic tools is essential to understand their function *in vivo* [1]. Spatiotemporal control of gene knockout has become the standard approach in vertebrate model organisms to investigate RNA and protein function. Multiple site-directed tools are available to control gene knockout, including recombinase systems such as Cre/LOX, Flp/FRT, and Dre/ROX, as well as endonuclease-mediated CRISPR editing [2,3]. However, these systems are not available in all vertebrate models, are sometimes challenging to implement, and can display variable efficiency depending on the genomic context. Hence, additional approaches are needed to control gene expression.

The synthetic biology field has contributed several tools to regulate RNA stability [4]. One promising family of RNA regulatory elements consists of self-cleaving ribozymes, which are short RNA structures capable of catalyzing their own cleavage [5]. When inserted in a protein-coding gene, this cleavage leads to RNA degradation and a reduction in protein levels. This approach has been used to modulate transgene stability in multiple models, including human and mouse cells [6–9], *C. elegans* [10,11], *Drosophila* [12], and *Plasmodium* [13]. One limitation to efficient RNA degradation is the slow cleavage kinetics, which makes most natural self-cleaving ribozymes unsuitable for *in vivo* applications. To circumvent this issue, an artificial *Schistosoma mansoni* hammerhead ribozyme, called *T3H48*, which displays a dramatic increase in cleavage rate has recently been engineered [14]. This ribozyme, as well as related ones [11], have been used to control transgene expression using Cre- and Flp-recombinases [15].

Although this approach has proven to be efficient in regulating exogenous constructs, it remains unclear whether engineered self-cleaving ribozymes can control endogenous gene expression in vertebrates. The zebrafish emerged several decades ago as an alternative and powerful vertebrate genetic model [16]. Most genetic tools involving site-directed recombinases and endonucleases have been successfully implemented in zebrafish [17–19]. Moreover, the zebrafish model allows the efficient screening of genetic tools because of its external fertilization and rapid early development.

Here, we use the zebrafish model to establish a self-cleaving ribozyme-based system to knock down gene expression *in vivo*. We first use a GFP reporter assay to assess the efficiency of several self-cleaving ribozymes in reducing pre-mRNA levels and find that the *T3H48* hammerhead ribozyme (*T3H48-HHR*), when placed in an intron, causes a 20-fold reduction in *eGFP* mRNA. Next, we show that inserting the *T3H48-HHR* in the second intron of the *albino(alb)/slc45a2* gene recapitulates the pigmentation phenotype of the *alb* mutant [20]. Using a base-editing strategy to inactivate the ribozyme, we show that this phenotype is reversible, illustrating the specificity of the approach. Furthermore, we created a Flippase- and Cre-inducible *T3H48-HHR* cassette called RiboFlip, which we also inserted in the *alb* gene. We show that RiboFlip induction also leads to a pigmentation phenotype similar to that of the mutant. Together, these new approaches broaden the genetic toolbox available for functional genomics in vertebrates.

## Results

### The *T3H48* ribozyme induces gene knockdown in zebrafish

The *T3H48-HHR* and related ribozymes catalyze efficient RNA cleavage in several organisms [9,11,14,15]. However, their efficiency remains untested in a vertebrate model. To do so, we first generated a transgenic cassette that contains two ubiquitous *ubb* promoters [21] controlling separately an *eGFP* containing a human β-globin (*hHBB*) intron [22] and *mCherry*, each followed by a triple SV40 3'UTR to terminate transcription (Fig 1a). We injected this transgene at the one-cell-stage and show that the expression of eGFP and mCherry colocalize perfectly (S1a Fig). Most ribozyme applications rely on a 3'UTR insertion in a transgene [5].

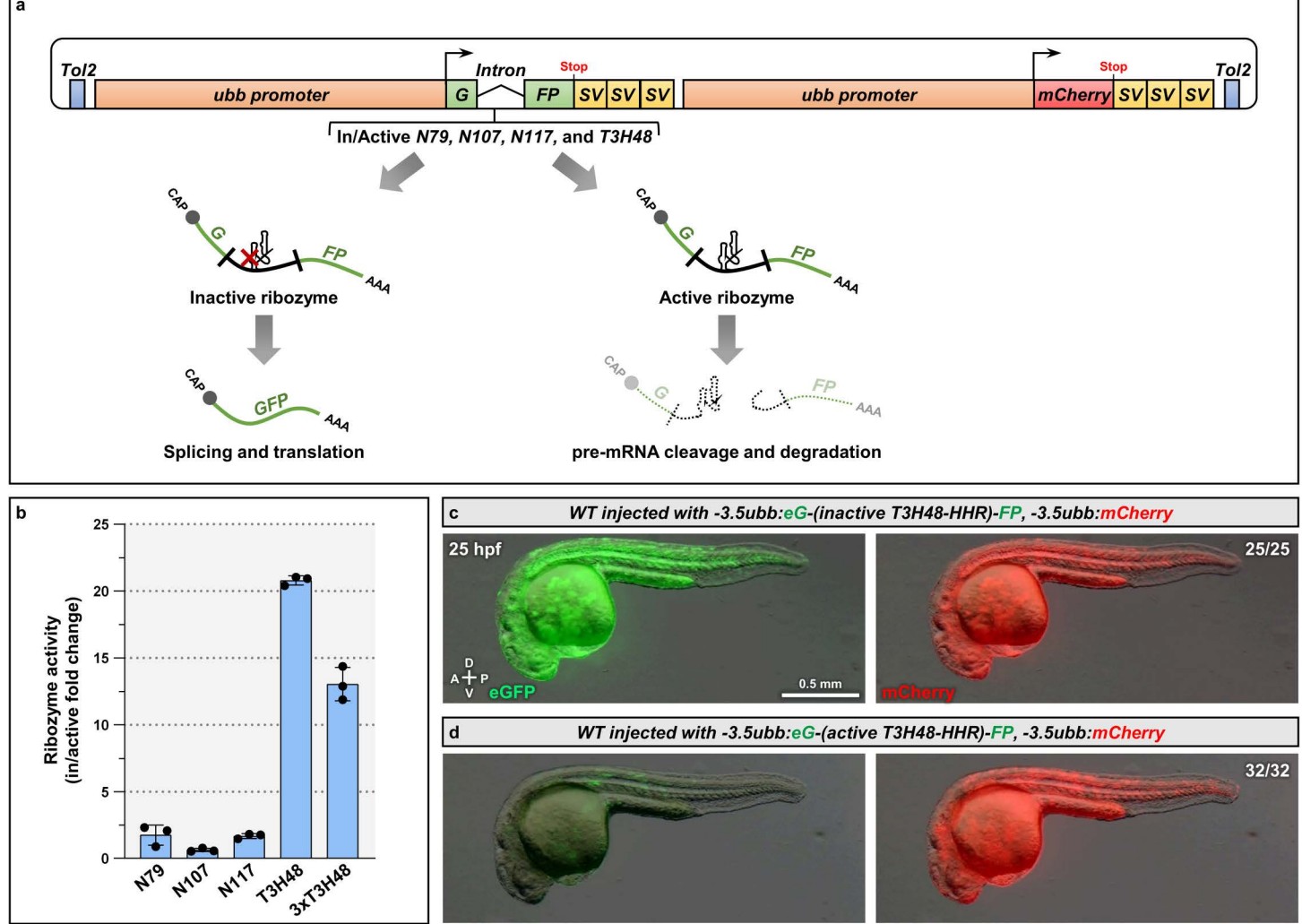

**Fig 1. The self-cleaving *T3H48* ribozyme induces pre-mRNA cleavage in zebrafish.** (a) Schematic of the dual *ubb* promoter vector that controls the expression of an *eGFP* containing a human β-globin (*hHBB*) intron and *mCherry*; the active and inactive ribozymes are inserted in the *hHBB* intron; red x indicates the location of the ribozyme inactivating mutation. (b) Activity of the *N79*, *N107*, *N117*, *T3H48*, and *3xT3H48* ribozymes represented as the ratio of *eGFP* mRNA levels in embryos injected with the active or inactive ribozyme construct; *mCherry* was used as a reference gene in this experiment to account for the variable number of plasmid copies in each injected embryo; n=3 biologically independent samples; Ct values are listed in S1 Table. (c-d) Merge of brightfield and fluorescence images of 25 hpf (hours post-fertilization) embryos injected at the one-cell stage with the dual *ubb* vector, comprising an inactive (c) or active (d) *T3H48-HHR*. The proportion of embryos matching the image shown is indicated in the top right corner of each image. The diagram indicates the Anterior-Posterior (A-P) and Dorsal-Ventral (D-V) axes.

To maintain the structure of the mature mRNA and increase the number of available landing sites for ribozyme integration, we inserted it in the intron of *eGFP*. We compared the engineered *T3H48* hammerhead ribozyme, its original counterpart, *N107*, and two other related ribozymes, *N79* and *N117* [6], along with their catalytically inactive versions [6,14] (Figs 1a and S1b). We used a triple CAAA insulating sequence [10] to isolate the secondary structure of all ribozymes from the neighboring sequences [12] (S1b Fig). We measured *eGFP* mRNA levels and used *mCherry* as a reference gene to account for the variable number of plasmid copies received after injection. We determined ribozyme activity by assessing the ratio of *eGFP* mRNA levels between active and inactive versions of each ribozyme. We found that the *T3H48-HHR* causes a 20-fold reduction in *eGFP* mRNA levels, in contrast to less than 3-fold for *N79*, *N107*, and *N117* (Fig 1b). We also assessed whether 3 copies of *T3H48-HHR*, all flanked by triple CAAA insulating sequences, would improve cleavage efficiency. In fact, we observed lower activity when compared with a single copy (Fig 1b). Moreover, we found that eGFP fluorescence intensity was strongly reduced in embryos injected with the active *T3H48-HHR* compared with those injected with the inactive version, in contrast to the mCherry signal, which was comparable between all injected embryos (Fig 1c,d). Altogether, these data show that the *T3H48-HHR* is an efficient ribozyme for gene knockdown in zebrafish.

## Ribozyme integration in an endogenous locus recapitulates the mutant phenotype

Robust reduction in transgenic mRNA levels after ribozyme cleavage has already been shown in other models [9,10,12]. Here we investigated whether integrating *T3H48-HHR* in an endogenous locus could induce phenotypes similar to those observed in the mutant. We targeted the *alb* gene, whose downregulation leads to strong pigmentation defects [20,23]. We integrated the active *T3H48-HHR* in the second intron of the *alb* gene using CRISPR/Cas9 knockin [18] using an already described CRISPR site [24] (Figs 2a and S2a). We flanked the ribozyme with a double CAA insulating sequence (S1b Fig), a shorter one than that used in Fig 1 in order to fit it into a 120 bp ssODN donor [18]. We refer to the obtained *Pt(alb:alb-T3H48-HHR)* allele as *alb^HHR* hereafter. We found that *alb^HHR/HHR* embryos display a 13.6-fold reduction in *alb* mRNA levels at 36 hpf (Fig 2b). This reduction appears to occur independently of splicing defects, as the *alb* mRNA displays a wild-type-like size in *alb^HHR/HHR* embryos as assessed by an RT-PCR reaction at saturation (S2b Fig). Moreover, *alb^HHR/HHR* and *alb^HHR/b4* embryos also display a pronounced decrease in pigmentation, similar to the *alb^b4/b4* phenotype (Fig 2c-f). The pigmentation phenotype then recovers in *alb^HHR/HHR* and *alb^HHR/b4* larvae (S2c–f Fig). However, this recovery is not due to a weakening of the cleavage efficiency of the ribozyme, as *alb* mRNA levels remain low in *alb^HHR/HHR* larvae (S2g Fig). In addition, we wanted to investigate whether the *T3H48-HHR* insertion, rather than its cleavage, could be responsible for the pigmentation phenotype. To this aim, we converted the active *T3H48-HHR* ribozyme insertion in the *alb* gene to an inactive version using the ABE-Ultramax (Umax) base editing strategy [25] (Fig 2g). We found that this conversion very efficiently rescued the pigmentation phenotype in *alb^HHR/b4* embryos (Fig 2h-k). Altogether, these data show that the *T3H48* ribozyme can induce the cleavage of an endogenous mRNA target in zebrafish.

## A Flippase- and Cre-activatable *T3H48* ribozyme allows conditional gene knockdown

Spatiotemporal control of the *T3H48-HHR* and related ribozymes has been achieved in *C. elegans* [11] and with the use of AAV vectors [15]. However, ribozyme activation *in vivo* in the endogenous locus of a vertebrate has never been attempted. Here, we designed a cassette

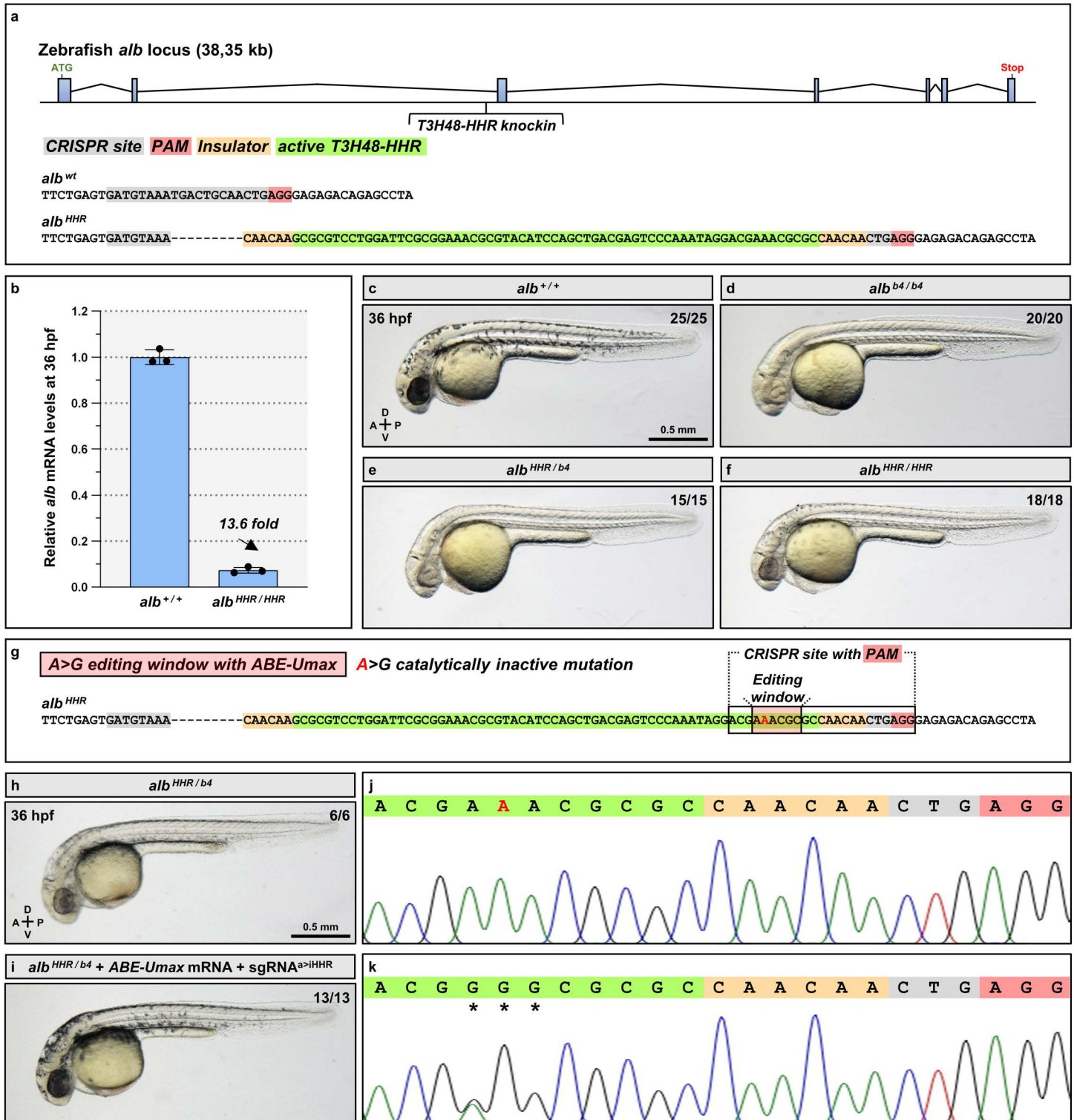

**Fig 2. Intronic insertion of the *T3H48* ribozyme in the *alb* gene recapitulates the mutant phenotype.** (a) Schematic of the *alb* locus showing the position of the *T3H48-HHR* integration, the target CRISPR site, and the genomic sequences before and after insertion. (b) Relative *alb* mRNA levels in 36 hpf wild-type and *alb*HHR/HHR embryos; n=3 biologically independent samples; Ct values are listed in S1 Table. (c-f) Brightfield images of 36 hpf wild-type (c), *alb*b4/b4 (d), *alb*HHR/b4 (e), and *alb*HHR/HHR (f) embryos. (g) Schematic of the integrated *T3H48-HHR* in the *alb* gene, the new CRISPR site created by the insertion, the A>G editing window of the ABE-Umax, and the A>G inactivating mutation in the ribozyme. (h-k) Brightfield imaging (h-i) and sanger sequencing (j-k) of part of the ribozyme and flanking

element from 36 hpf *alb*HHR/b4 non-injected (h,j) or injected with *ABE-Umax* mRNA and a sgRNA that inserts an inactivating mutation in the *T3H48-HHR* (i,k). The proportion of embryos matching the image shown is indicated in the top right corner of each image. The diagram indicates the Anterior-Posterior (A-P) and Dorsal-Ventral (D-V) axes.

called RiboFlip, that allows the activation of *T3H48-HHR* using Flippase (Flp), and the dual activation of *T3H48-HHR* and a *5XUAS:TagBFP* using Cre (Figs 3a and S3a). We strategically positioned two different FRT and LOX sites in a head-to-head orientation. This positioning allows the stable inversion of the cassette after Flp or Cre recombination: a first round of recombination will invert the cassette and a second round will excise one recombination site, leaving heterotypic sites on both sides unable to recombine [17,26–28]. We inserted this cassette in the same intron of the *alb* gene as in Fig 2a, using CRISPR/Cas9 knockin [29]. We refer to the obtained *Pt(alb:alb-RiboFlip-OFF)^bns697* allele as *alb^R-OFF* hereafter. We found that both Flp- and Cre-activation of *alb^R-OFF* lead to the flipping of the cassette (S4a Fig), *alb* mRNA knockdown (Fig 3b), and pigmentation defects (Fig 3c-e). We refer to the stable *Pt(alb:alb-RiboFlip-Flp-ON)^bns732* and *Pt(alb:alb-RiboFlip-Cre-ON)^bns733* alleles as *alb^R-Flp-ON* and *alb^R-Cre-ON* respectively hereafter. Conversely, we observed a rescue of the pigmentation phenotype when *alb^R-Flp-ON/R-Flp-ON* embryos were injected with *Cre* mRNA at the one-cell-stage (Fig 3f). Importantly, we found that these phenotypes were not due to mRNA perturbation as the *alb* mRNA displays a wild-type-like size after RiboFlip insertion as assessed by an RT-PCR reaction at saturation (S2b and S4b Figs). We also included universal CRISPR/*SpyCas9* and CRISPR/*Lba-Cas12a* sites on the RiboFlip cassette (Figs 3a and S3a), so that it can be targeted for knockout (S4c Fig) or used as a landing site for the additional knockin of an *mRFP* reporter (S4d,e Fig). Current systems reporting recombination are only visible when the target gene is expressed [17,30,31], or by an independent transgene [32,33]. We then assessed whether Cre recombination could be visualized regardless of *alb* expression pattern using RiboFlip. We show that Cre activation of the *alb^R-OFF* allele can be reported using the GAL4/UAS system, after *Gal4* mRNA injection at the one-cell-stage (S4f-g Fig), and turned off using the Dre/ROX system, which excises the *TagBFP* reporter (Figs 3a,S3a and S4h). Thus, this reporter system allows the visualization of Cre-recombined cells independently of target gene expression, as shown with a *Gal4* expressed under the control of the *myl7* cardiomyocyte regulatory element [34] (S4i Fig), a cell type that does not express *alb* [23]. Taken together, our data show that RiboFlip is flexible conditional knockdown system.

## Discussion

Self-cleaving ribozymes are already used to control the expression of transgenes and endogenous genes in multiple models [6–8,10–15]. However, modulating the expression of an endogenous locus in a vertebrate model has never been achieved. In our study, we used *T3H48-HHR* in zebrafish to modulate the mRNA levels produced from a reporter transgene as well as the *alb* gene. We also used CRISPR/Cas9 knockin strategies to insert an inducible *T3H48-HHR* cassette called RiboFlip and show that this system can trigger conditional gene knockdown of the endogenous *alb* gene.

Ribozymes are usually inserted in UTR regions to mediate efficient knockdown throughout the lifetime of mRNAs. As the pre-mRNA is accessible only before splicing, intronic insertions induce lower cleavage rates [6]. However, we reasoned that the improved cleavage rate of *T3H48-HHR* could induce efficient pre-mRNA cleavage. Intronic cleavage could increase the number of potential landing sites to insert the ribozyme and avoid the destabilization of the mRNA when fused with a drug-inducible aptamer [10]. We first assessed the efficiency of *T3H48-HHR* in mediating pre-mRNA cleavage using a reporter strategy. Although this

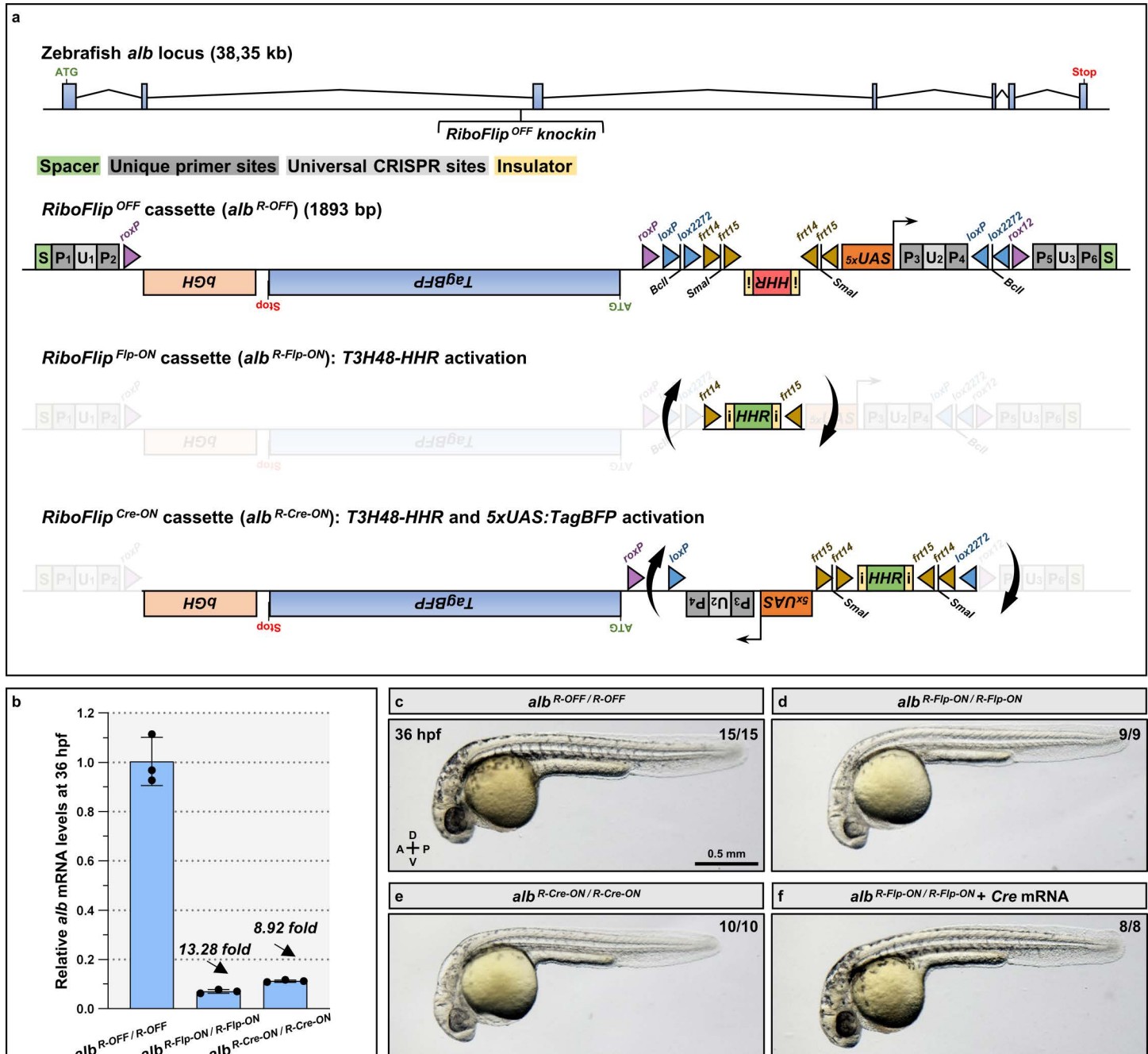

**Fig 3. RiboFlip is a Flp- and Cre-inducible knockdown cassette.** (a) Schematics of the *alb* locus showing the position of the RiboFlip integration and the details of the cassette in the *T3H48-HHR OFF* orientation, *Flp-ON* orientation, and *Cre-ON* orientation; *T3H48-HHR* antisense orientation is in red and sense orientation in green; RiboFlip includes six unique primer sites (*P1-6*), three universal CRISPR sites (*U1-3*), a β-globin terminator (*bGH*) downstream of the *TagBFP*, and recombination sites (*LOX/FRT/ROX*). (b) Relative *alb* mRNA levels in 36 hpf *alb*R-OFF/R-OFF, *alb*R-Flp-ON/R-Flp-ON, and *alb*R-Cre-ON/R-Cre-ON embryos; n=3 biologically independent samples; Ct values are listed in S1 Table. (c-f) Brightfield images of 36 hpf *alb*R-OFF/R-OFF (c), *alb*R-Flp-ON/R-Flp-ON (d,f), and *alb*R-Cre-ON/R-Cre-ON (e) embryos non-injected (c-e) or injected at the one-cell stage with *Cre* mRNA (f). The proportion of embryos matching the image shown is indicated in the top right corner of each image. The diagram indicates the Anterior-Posterior (A-P) and Dorsal-Ventral (D-V) axes.

approach led to lower cleavage rates than expected [14], it induced a robust loss of reporter expression. Further work should assess whether UTR cleavage leads to stronger degradation in vertebrate models *in vivo*. Surprisingly, the insertion of three copies of *T3H48-HHR* did not increase the cleavage rate. We hypothesize that the secondary structure of the ribozyme, which is essential for cleavage, was altered by this array. Additional insulating sequences [12] should be assessed to maintain a high cleavage rate in ribozyme arrays.

We then evaluated the ability of *T3H48-HHR* to recapitulate mutant phenotypes using *alb* as a model. We inserted the ribozyme in the second intron of the *alb* gene, in a CRISPR site previously used to induce large deletions [24]. We observed a weaker reduction in *alb* mRNA levels compared with the reporter experiment. This difference could be due to a positional effect of the ribozyme, whose folding depends on the neighboring sequences [12], or by the shorter insulating sequence used in this experiment to reduce the size of the ssODN donor for knockin. Notably, the larvae, unlike the embryos, started exhibiting wild-type-like pigmentation. The *alb*^b4 mutants lack almost all melanin but display normal melanosomes [23]. Considering the high stability of melanin [35], the recovery in *alb*^HHR/HHR larvae is likely due to Alb protein accumulation, since the ribozyme cleavage efficiency did not appear to diminish over time (S2g Fig). Several hypomorphic *alb* alleles have already been reported [23]. However, in the absence of a mutant allele showing reduced *alb* expression without lesions in the coding sequence, this explanation remains hypothetical and should be tested. Moreover, hypomorphic phenotypes generated by ribozyme cleavage could be used to investigate genes for which the knockout results in early lethality.

We then showed that it is indeed the ribozyme cleavage and not the ribozyme insertion that is responsible for the pigmentation phenotype. We inactivated the *T3H48* ribozyme integrated in the *alb* gene using a base editing strategy. We propose this approach as an new method to turn off the activity of HHR ribozymes *in vivo*, notably as an alternative to aptamers designed to change the conformation of self-cleaving ribozymes upon drug binding [8,10].

The *T3H48-HHR* ribozyme is an RNA structure that catalyzes its own cleavage. We reasoned that integrating the ribozyme in an antisense orientation would prevent its cleavage, as was recently shown in a transgenic setting [15]. To that end, we created a RiboFlip cassette that switches *T3H48-HHR* to a sense orientation following Flippase or Cre recombination and inserted it in the second intron of the *alb* gene. Recombination triggers mRNA knockdown and a pigmentation phenotype similar to the effects caused by of *T3H48-HHR* insertion. The Cre recombination induced a weaker knockdown than the Flippase one, possibly due to the difference in the sequence flanking the ribozyme, which could alter folding. Additionally, Cre-induced recombination connects a *UAS* with a *TagBFP* to label the recombined cells independently of the target locus. Although TagBFP expression became visible after Gal4 expression, the signal was weak and mosaic, possibly due to a suboptimal design, as a large sequence separates the *UAS* from the *TagBFP*, or to the genomic context in the *alb* gene. For future studies, a short ubiquitous promoter would probably be preferable to the UAS, which requires the use of an additional Gal4 line.

We also included in the RiboFlip cassette multiple universal CRISPR sites that can be used to turn RiboFlip into a landing site for additional knockins. Furthermore, we integrated in the RiboFlip cassette heterotypic ROXP or ROX12 sites [36]; these sites can be used to remove the remaining RiboFlip elements after knocking in another cassette containing a ROX site. We anticipate that these universal CRISPR sites could be used as alternative landing sites to the existing *attP* used for phiC31-mediated knockin [37,38].

In zebrafish, the RNA knockdown toolset includes shRNAs [39], microRNAs [40,41], and CRISPR-based RNA editors [42–44]. However, their efficiency and specificity could be improved. Since ribozyme-mediated knockdown requires ribozyme integration into the target

locus, it provides very high specificity. Moreover, the mRNA knockdown rate of more than 90% that we observed *in vivo* with the GFP reporter system and in the *alb* locus has never been achieved with other tools.

In summary, our findings indicate that *T3H48-HHR* can mediate rapid gene knockdown in a vertebrate model when inserted in an intron. Moreover, this ribozyme can be induced conditionally using recombinases and represents a flexible alternative to existing knock-down strategies.

## Methods

### Ethics Statement

All procedures performed on animals conform to the guidelines from Directive 2010/63/EU of the European Parliament on the protection of animals used for scientific purposes and were approved by the Animal Protection Committee (Tierschutzkommission) of the Regierungspräsidium Darmstadt (reference: B2/1218).

### Zebrafish husbandry

Zebrafish husbandry was performed under institutional (MPG) and national (German) ethical and animal welfare regulations. Larvae were raised under standard conditions. Adult zebrafish were maintained in 3,5 l tanks at a stock density of 10 zebrafish/l with the following parameters: water temperature: 27–27,5°C; light/dark cycle: 14/10; pH: 7,0–7,5; conductivity: 750–800 μS/cm. Zebrafish were fed 3–5 times a day, depending on age, with granular and live food (*Artemia salina*). Health monitoring was performed at least once a year. All embryos and larvae used in this study were raised at 28°C and staged at 75% epiboly for synchronization.

### Knockin strategies

*T3H48-HHR* and RiboFlip were inserted in the same CRISPR site in the *alb* gene, located 312 bp away from the third exon [24], using CRISPR/Cas9 knockin strategies. *T3H48-HHR* and its insulators were inserted using an unmodified ssODN (Sigma) and 21 bp symmetric homology arms [18], for a total length of 119 bp. RiboFlip (Fig 3a) and the mRFP U-CRISPR reporter (S4d Fig) were inserted using 5'AmC6-modified (Sigma) PCR products [29], which includes 48 bp symmetric homology arms [45]. The RiboFlip cassette was *de novo* synthesized and cloned into a pUC57 vector (GenScript). The RiboFlip donor PCR product was amplified from the pUC57 vector using the P1/P6 primer pair, including the spacers (S3a Fig). The *mRFP* reporter, together with a branching point/splice acceptor (*BP/SA*), a *P2A* peptide, and an *Ocean Pout* terminator was amplified from pUFlip-floxed2A-mRFP-; gcry1:BFP -1, a gift from Jeffrey Essner (Addgene plasmid # 173887) [17], using a flanking primer pair. All the primers used for the knockin and the ribozyme donor are available in S1 Table.

### Zebrafish strains and genotyping protocol

The following lines were used in this study: *alb^b4* [20], *Tg(myl7:GAL4)^cbg2* [34], *Tg(hsp70l:Cre)^zdf13* [46]. The following lines were newly generated: *Pt(alb:alb-T3H48-HHR)^bns696* (or *alb^HHR*), *Pt(alb:alb-RiboFlip-OFF)^bns697* (or *alb^R-OFF*), *Pt(alb:alb-RiboFlip-Flp-ON)^bns732* (or *alb^R-Flp-ON*), *Pt(alb:alb-RiboFlip-Cre-ON)^bns733* (or *alb^R-Cre-ON*).

For the experiments with the *T3H48-HHR* knockin, we intercrossed *alb^HHR/+* (Figs 2a and S2a,b,g) or *alb^HHR/b4* (Figs 2c–f and S2c–f) parents, or crossed *alb^HHR/+* with *alb^b4/b4* parents (Fig 2h-k) and genotyped *alb^+/+*, *alb^b4/b4*, *alb^HHR/b4*, and *alb^HHR/HHR* embryos using PCR primers that allow one to distinguish between all alleles (S2a Fig). We used heterozygous parents, as

*alb* mRNA is not maternally provided [47], and there are no phenotypic differences between zygotic mutants and maternal-zygotic mutants.

For the experiments with the *RiboFlip* knockin, we intercrossed $alb^{R-OFF/+}$, $alb^{R-Flp-ON/+}$, or $alb^{R-Cre-ON/+}$ parents and genotyped embryos/larvae using wild-type and knockin-specific PCR (S4a Fig for the Flp- and Cre-recombined alleles).

For the Dre/ROX experiment, the recombination was assessed using roxP flanking primers.

For the *myl7:Gal4* experiments, the Gal4 was genotyped with an allele-specific PCR.

PCR amplifications were performed with KAPA2G Fast Ready Mix (Sigma 2GFRMKB).

For the T7 endonuclease I (NEB M0302) assay [48], we extracted DNA from a pool of 5 embryos injected with SpyCas9 or LbaCas12a proteins (IDT), complexed with universal sgRNAs (Cas9) or crRNAs (Cas12) (S3a, S4c Figs), *in vitro* synthesized (Invitrogen AM1354) using T7 promoters [49]. All the primers used to genotype the alleles described in this study are listed in S1 Table.

## Plasmid generation and injections

The following plasmids were assembled using *in vivo* cloning [50]: *-3.5ubb:eG-(Ribozymes)-FP*, *-3.5ubb:mCherry*. Two *ubb* ubiquitous promoters [21] control the expression of an *eGFP* gene containing a *hHBB* intron [22] and an *mCherry* gene, both terminated by a triple SV40 late polyA sequence. We cloned the active and inactive versions of the *N79*, *N107*, *N117* [6], *T3H48* [14], and *3xT3H48* ribozymes in the *hHBB* intron. All ribozymes are flanked by a triple CAAA insulator sequence [10], except the knockin in Fig 2a, which is flanked by a double CAA insulator, which we modified to fit into a short 120 bp ssODN [18].

The *Flp°* [51] and *Dre* [52] were zebrafish-codon optimized using the iCodon algorithm [53], synthesized *de novo* (GeneScript), cloned into a pT3TS vector, linearized using KpnI, and *in vitro* synthesized using the mMessage mMACHINE T3 transcription kit (Invitrogen AM1348). pCS2-*Cre.zf1*, a gift from Harold Burgess (Addgene plasmid # 61391) [54], and pCS2+-Gal4FF [55] were linearized using NotI and *in vitro* synthesized using the mMessage mMACHINE SP6 transcription kit (Invitrogen AM1340). pT3TS-spCas9 ABE-Umax was a gift from Gaurav Varshney (Addgene plasmid # 222138) [25], and base editing was performed according to a published protocol [25] using an Alt-R-modified sgRNA by IDT.

We engineered the locking of *T3H48-HHR* in the sense orientation using a combination of two hetero-specific FRT14/FRT15 and LOXP/LOX2272 recombination sites, similar to a recent strategy [15]. We injected 20 pg of *Flp°* mRNA to establish the $alb^{R-Flp-ON}$ line. We injected 6 pg of *Cre* mRNA to establish the $alb^{R-Cre-ON}$ line and also for transient experiments (Fig 3f). We injected 10 pg of *Gal4* mRNA for *UAS* induction in transient experiments (S4d-f Fig); in these embryos, we injected 6 pg of *Dre* mRNA to remove the *TagBFP* reporter (S4f Fig). We injected 15 pg of plasmid DNA for transient reporter experiments. All reagents were injected at the one-cell stage in a volume of 1 nl together with 0,2% phenol red.

## PCR and RT-qPCR analysis

PCR reactions on genomic DNA and cDNA to visualize $alb^{HHR}$, $alb^{R-OFF}$, $alb^{R-Flp-ON}$, and $alb^{R-Cre-ON}$ integration (S2a Fig), recombination (S4a Fig), and mature RNA integrity (S2b,S4b Figs) were performed on single embryos using KAPA2G Fast Ready Mix (Sigma 2GFRMKB) (S2a, S4a Figs) and PrimeSTAR Max DNA Polymerase (Takara) (S2b,S4b Figs). A nested PrimeSTAR was necessary to visualize *alb* cDNA.

RT-qPCR analysis was performed on pools of 5 zebrafish collected at 25 (Fig 1b),36 (Figs 2b,3b), and 72 (S2g Fig) hpf, and the results represent biological triplicates with two technical duplicates per biological replicate. *mCherry* (Fig 1b) and *eef1b2* (Figs 2a,S2g and 3b) were

used as reference genes for the RT-qPCR data analysis, and fold changes were calculated using the 2−ΔΔCt method. RNA was isolated using TRIzol extraction and reverse transcription was performed using Maxima First Strand cDNA synthesis (Thermo Fisher K1641) for PCR on cDNA and RT-qPCR experiments. Embryos were PCR genotyped before RNA extraction for PCR on cDNA and RT-qPCR experiments by cutting a piece of the tail. All Ct values and primers are listed in S1 Table.

### Brightfield and fluorescence imaging

We performed brightfield and fluorescence imaging of whole embryos with a pixel width of 3,24 µm using a SMZ25 stereomicroscope (Nikon) with a 2x/0.3 objective (Figs 1c–d, S1a, 2c–f, S2c–f,3c–f and S4d–f). Heart images were acquired using an LSM 700 confocal laser scanning microscope (Zeiss) using a 40x objective with a pixel width of 0,52 µm and a z-step of 2 µm (S4g Fig).

### Statistics and Reproducibility

Data were processed with GraphPad Prism 9 and Microsoft Excel 2016. Experiments were performed at least three times independently and only included when showing consistent results.

### Supporting information

**S1 Fig. eGFP and mCherry fluorescence colocalize under the control of dual ubb promoters.** (a) Merge of brightfield and fluorescence images of 25 hpf embryos injected at the one-cell stage with the dual *ubb* vector; the proportion of embryos matching the image shown is indicated in the top right corner. (b) Sequences of the *N79*, *N107*, *N107*, and *T3H48* hammerhead ribozymes used in this study and of the flanking insulators, and location of the inactivating mutation.
(TIF)

**S2 Fig. Phenotypic expressivity after *T3H48* ribozyme insertion does not depend on mRNA levels.** (a) Schematic of the *alb* locus and agarose gel image showing a PCR amplification of the *alb^HHR* region from 36 hpf wild-type, *alb^HHR/+*, and *alb^HHR/HHR* embryos. (b) Schematic of the *alb* mRNA and agarose gel image showing an RT-PCR amplification of the *alb* full-length mRNA from 36 hpf wild-type and *alb^HHR/HHR* embryos. (c-f) Brightfield images of 72 hpf wild-type (c), *alb^b4/b4* (d), *alb^HHR/b4* (e), and *alb^HHR/HHR* (f) larvae. (g) Relative *alb* mRNA levels in 72 hpf wild-type and *alb^HHR/HHR* larvae; n=3 biologically independent samples; Ct values are listed in S1 Table. The proportion of larvae matching the image shown is indicated in the top right corner of each image.
(TIF)

**S3 Fig. Description of the RiboFlip cassette.** (a) Annotated sequence of the RiboFlip cassette containing an extensive list of all components, including six unique primer sites (P1-6), three universal CRISPR sites (U1-3), a β-globin terminator (bGH) downstream of the TagBFP, and recombination sites (LOX/FRT/ROX).
(TIF)

**S4 Fig. The components of the RiboFlip cassette are functional.** (a) Schematics and agarose gel images of the RiboFlip cassette inserted in the *alb* locus showing a Flp and Cre flipping-specific PCR amplification of the indicated fragments from 36 hpf *alb^R-OFF/+*, *alb^R-Flp-ON/+*, and *alb^R-Cre-ON/+* embryos (left panel), and from *alb^R-OFF/+; hsp70l:Cre* embryos without

or with heat shock (HS) (right panel); the purple boxes represent the portion of the RiboFlip cassette that gets flipped upon Flp- or Cre-induction; heat shock was performed by placing 24 hpf embryos in pre-heated egg water at 39°C for 1 h three times in a row, spaced by 1 h at 28°C; two different primer pairs are used to visualize *Cre* mRNA- and *hsp70l:Cre*-mediated recombination. (b) Schematic and agarose gel image of the *alb* mRNA showing a RT-PCR amplification of *alb* full-length mRNA from 36 hpf *alb*$^{R-OFF/R-OFF}$, *alb*$^{R-Flp-ON/R-Flp-ON}$, and *alb*$^{R-Cre-ON/R-Cre-ON}$ embryos. (c) Schematic and sequence of the RiboFlip cassette inserted in the *alb* gene and the universal SpyCas9/LbaCas12 CRISPR sites (U-CRISPR), and agarose gel image of the T7 endonuclease I (T7EI) assay performed on 48 hpf *alb*$^{R-OFF/R-OFF}$ embryos; asterisks (*) highlight the degradation products of the T7EI assay. (d) Donor cassette schematic and fluorescence image of a 26 hpf *alb*$^{R-OFF/+}$ embryo injected at the one-cell stage with Cas9 protein, a U-CRISPR-targeting sgRNA, and a 5'AmC6-modified donor PCR product consisting of a branch point/splice acceptor (BP/SA) sequence, a P2A peptide, a mRFP reporter, and an ocean pout (OP) terminator (U-CRISPR mix). (e) Proportion of positive embryos expressing mRFP in retinal pigmented epithelial cells following injection with U1, U2, and U3 SpyCas9 U-CRISPR mix. (f-h) Fluorescence images of 25 hpf *alb*$^{R-OFF}$ (f) and *alb*$^{R-Cre-ON/+}$ (g,h) embryos injected at the one-cell stage with *Gal4* (f,g) or *Gal4* and *Dre* (h) mRNA. (i) Confocal images of hearts (ventral views) from 72 hpf *alb*$^{R-Cre-ON/+}$ and *alb*$^{R-Cre-ON/+}$; *myl7:GAL4*$^{+/-}$ larvae; maximum z-projection; annotations correspond to the heart ventricle (V) and atrium (A). The proportion of embryos and larvae matching the image shown is indicated in the top right corner of each image. The diagrams indicate the Anterior-Posterior (A-P), Dorsal-Ventral (D-V), and Left-Right (L-R) axes.
(TIF)

**S1 Table. CRISPR sites, donors, primers, and Ct values.**
(PDF)

## Acknowledgments

We thank Chris SM Helker for providing the *pCS2+-Gal4FF* vector and Kenny Mattonet for designing the flipping FRT sites.

## Author contributions

**Conceptualization:** Thomas Juan, Didier Y.R. Stainier.

**Data curation:** Thomas Juan, Tonatiuh Molina, Sofia Papadopoulou, Bárbara Cardoso, Shivam Govind Jha, Didier Y.R. Stainier.

**Formal analysis:** Thomas Juan, Tonatiuh Molina, Lihan Xie, Sofia Papadopoulou, Bárbara Cardoso, Shivam Govind Jha, Didier Y.R. Stainier.

**Funding acquisition:** Thomas Juan, Didier Y.R. Stainier.

**Investigation:** Thomas Juan, Tonatiuh Molina.

**Methodology:** Thomas Juan, Tonatiuh Molina, Lihan Xie, Didier Y.R. Stainier.

**Project administration:** Thomas Juan, Didier Y.R. Stainier.

**Resources:** Thomas Juan, Didier Y.R. Stainier.

**Software:** Thomas Juan, Didier Y.R. Stainier.

**Supervision:** Thomas Juan, Didier Y.R. Stainier.

**Validation:** Thomas Juan, Tonatiuh Molina, Lihan Xie, Sofia Papadopoulou, Bárbara Cardoso, Shivam Govind Jha, Didier Y.R. Stainier.

**Visualization:** Thomas Juan, Tonatiuh Molina.

**Writing – original draft:** Thomas Juan, Tonatiuh Molina, Didier Y.R. Stainier.

**Writing – review & editing:** Thomas Juan, Tonatiuh Molina, Lihan Xie, Sofia Papadopoulou, Bárbara Cardoso, Shivam Govind Jha, Didier Y.R. Stainier.

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
