## [Decision Letter · Decision Letter 0]

19 Sep 2024

Dear Dr Stainier,

Thank you very much for submitting your Research Article entitled 'A recombinase-activated ribozyme to knock down endogenous gene expression in zebrafish.' to PLOS Genetics.

The manuscript was fully evaluated at the editorial level and by independent peer reviewers. The reviewers appreciated the attention to an important problem, but raised some substantial concerns about the current manuscript. Based on the reviews, we will not be able to accept this version of the manuscript, but we would be willing to review a much-revised version. We cannot, of course, promise publication at that time.

If you decide to revise the manuscript for further consideration at PLOS Genetics, please aim to resubmit within the next 60 days, unless it will take extra time to address the concerns of the reviewers, in which case we would appreciate an expected resubmission date by email to plosgenetics@plos.org.

If present, accompanying reviewer attachments are included with this email; please notify the journal office if any appear to be missing. They will also be available for download from the link below. You can use this link to log into the system when you are ready to submit a revised version, having first consulted our Submission Checklist .

PLOS has incorporated Similarity Check , powered by iThenticate, into its journal-wide submission system in order to screen submitted content for originality before publication. Each PLOS journal undertakes screening on a proportion of submitted articles. You will be contacted if needed following the screening process.

To resubmit, log into your Editorial Manager account and select the option 'Revise Submission' in the 'Submissions Needing Revision' folder.

We are sorry that we cannot be more positive about your manuscript at this stage. Please do not hesitate to contact us if you have any concerns or questions.

Yours sincerely,

Jeffrey J Essner

Guest Editor

PLOS Genetics

Aleksandra Trifunovic

Section Editor

PLOS Genetics

While the manuscript represents a good first start for using ribozymes for conditional genetics, the demonstrated efficacy is low. The knockdown efficacy is incomplete and may complicate the systems use for conditional genetics. Given the turnaround time for revision, it will not likely be feasible to make new lines for analysis. Please address the reviewers comments prior to publication as best you can.

Reviewer's Responses to Questions

**Comments to the Authors:**

Reviewer #1: The manuscript by Juan et al. describes the use of a ribozyme to knockdown endogenous gene expression in zebrafish. The strategy is elegant in that the ribozyme sequence is integrated into the intron of using CRISPR/Cas9 gene editing so that the preRNA is degraded following transcription. Several different ribozymes were tested for this purpose to determine the most efficient system. This represents the first time such a system has been used in vertebrates and has the potential to be a useful genetic tool. The efficiency of the system has the potential to induce phenotypic changes and displays approximately 13-fold knockdown of the golden transcript. However, to make the system of broad utility greater knockdown would be needed, as even at the golden locus, the phenotype recovers over time. This should not detract the achievement from the applying this for the first time in vertebrates, but it is highly recommended that the language is softened and discussion of the degree of knockdown for efficient application is discussed. For example, as little as 5% of activity for many genes is sufficient to not produce a phenotype.

The system is then used with recombinase systems to flip the ribozyme so that it is no longer expressed on the RNA level in the correct orientation. This inactivates the ribozyme and allows the system to be used for conditional genetics. This is a step forward for conditional genetics, but the incomplete knockdown may make phenotypic analysis complicated. Nevertheless, this would be the end goal of integrating the ribozyme into an intron. Overall, the manuscript represents an important step forward for using ribozymes for conditional genetics and a platform to improve knockdown to the point where the system would have widespread use.

Please address the following minor points prior to publication.

Soften the language for this being an efficient system throughout the manuscript.

Throughout the results, the conclusions of the experiment are often stated without describing the rationale or the result itself leaving the reader to try to fill in the blanks. For example, it is simply stated that “that the reduction occurs independently of splicing defects.” This is just one example of many instances where the results section could be expanded to help the reader. It also recommended the strategy for using different loxp and frt sites be explained. Having outcomes/predictions following recombinase delivery (especially because this is different for different recombinases) will be important for most readers.

Please define all abbreviations used in the figures in the figure legends.

Reviewer #2: Juan et al. examined the utility of T3H48 ribozyme for conditional gene knockdown in zebrafish. The authors demonstrate that this ribozyme can inactivate the expression of an endogenous gene when inserted in an intron of the target gene. They created an invertible cassette called RiboFlip, which does not suppress gene expression when inserted in the reverse orientation but does so only after the Cre- or Flp-mediated cassette inversion. The genetic approach for the conditional inactivation of gene expression is still limited in zebrafish, and the approach described in this study will be a valuable addition to the genetic toolbox available in the community. I have several comments and questions about the study, as described below:

1) I like the study, but I am a bit concerned about the lack of a convincing example of a loss-of-function phenotype achieved with the presented method. The pigments eventually return in alb-HHR/HHR (Fig. S2e, f). Is such a phenotype typical for the loss-of-function experiment with the alb gene?

2) I am confused with Fig. S2b. How could alb-HHR/HHR embryos show fewer pigments at 36 hpf (Fig. 2f), although they seem to have mature alb-mRNA? (Fig. S2b).

3) The purpose of using the insulator may need more explanation. It would also be appreciated if the authors could provide more details about the insulator used in this study, as it is not the one commonly used in transgenic constructs.

4) Does this method degrade mRNA through NMD and cause genetic compensation?

5) Fig. 3a, RiboFlip-Cre-ON panel: It might be better to remove the shade over TagBFP to make it clearer that Cre-mediated recombination induces TagBFP expression under the control of the UAS promoter.

6) Please clarify the labeling in Fig. 3b: R-OFF-/-, R-Flp-ON-/-, and R-Cre-ON-/-.

7) It would be better to include positive controls in Fig. S4a.

8) It would be helpful if major base pair sizes were indicated with the DNA ladders.

Reviewer #3: The manuscript of Juan et al. describes the use of a ribozyme to perform gene Knock-Down (KD) in Zebrafish, a vertebrate model.

The manuscript is direct and clear, and the Zebrafish and other vertebrate communities can benefit to the implementation of this new molecular tool for advanced genetic manipulations. Although I appreciate that the manuscript is direct, it could be really strengthened by providing more data and applications

Main comments:

-While the characterization of various ribozymes on a dual coding transgene (eGFP/mCherry) is a great starting point, in vivo applications are limited to the albino gene. It seems that the manuscript will be stronger and could lead to a wide adoption of the ribozyme if the authors use their system on other genes, and not be limited to such a ‘generic gene’. Describing, even superficially, some new biological functions by KD several endogenous genes will be beneficial and will increase the impact of this study.

-The authors propose ribozyme-KD as a way to extend the set of tools for genetic manipulation in Zebrafish and possibly in other vertebrate models. Therefore, it would be interesting to describe other tools available in Zebrafish and vertebrate in general (eg RNAi, shRNA) and describe why a ribozyme could be a unique tool for gene KD.

-Since one of the usefulness of a ribozyme is to KD genes, the authors should use it to KD a gene for which KO allele cannot be obtained because mutants are not viable.

-The authors state that their system is conditional. While true, injecting Flp or Cre is still cumbersome. It would be interesting to have these enzymes under the control of a heat-shock promoter to make the tool versatile and easy to use for the community. Also, I am not sure of what the authors mean by the advantage of “the visualization of Cre recombined cells independently of the targeted gene expression”, with the example of the cardiomyocyte regulatory element. This is because always 100% of the Cre injected embryos are not mosaic for recombination? Because having the recombination marker active in the same cells that express the gene targeted could be practical, since it would show that for sure the ribozyme is active in these cells. Moreover, having mosaic recombination with same cell marker can be better, as commonly done in Drosophila and Flp/ Heat-Shock for cell autonomous effect.

-The authors could try, instead of three ribozyme copies in the same intron (which is less efficient than a single ribozyme copy), to put several ribozyme copies spread on different introns. If such strategy improves the KD, this could really justify using ribozymes over other KD strategies… For instance, the authors could study a gene with a dose response phenotype.

Minor comments:

-The sentence “eGFP split by a human B-globin intron” might be confusing in regard of the split GFP system. Maybe something like “eGFP coding sequence containing an intron” could be use…

-Only in the discussion the recovering of pigmentation is explained, this should be done in the result part. How the authors rule out loss of ribozyme activity overtime or during some developmental stage?

-The discussion seems redundant with other manuscript part, with sometime very similar sentences.

-Insulators are mentioned in material and methods and figures, but not explained in the main text.

-Does the insulator and ribozyme insertion could affect expression of the targeted gene and neighbor genes? Use of the inactive ribozyme and quantitative PCR or other methods could be performed.

-Fig S2b legend: RT-PCR and not just PCR amplification?

-Not sure if calling the mCherry transgene a housekeeping gene is correct. It is more a control or reference gene.

-The Tol2 transgene is a verified single copy?

-Fig1a: the 3xSV40 utr in 3 yellow box is confusing since it looks like there are small introns inside. Why using 3 copies?

-Fig1a: a schematic of how ribozyme works could be very helpful for readers.

-Fig2a: the fact that the exogenous CRISPR sites in the sequence part are in light blue, as the exons in the gene model, is confusing.

-Ribozyme testing in homozygous lines should be better explained. How the crosses have been made, from homozygous parents, or Het providing wildtype maternal mRNA?

-For non-zebrafish readers, the genotype can be not clear in Fig4. It looks like it is for both background one allele with RiboFlip active (R-Flp-On or R-Cre-ON), and the other allele corresponds to two different mutant alleles (bns732 and bns733). Some precision in the legend could be useful for a broader readership.

-Some explanation about the difference between roxP-12, frt14/15, and loxP/2272 in Fig3a could also be useful

-Introducing exogenous/universal CRISPR sites in the ribozymes site is a good idea (FigS4), but the authors should show a downstream application by performing an in vivo experiment on them.

-FigS4a, could be nice to show where is the flipped sequence in the OFF version before Cre and Flp activation.

-FigS2b: so the mRNA level is reduced but the gel shows no reduction? This is confusing. Is it because the RT-PCR was saturated? If so, a RT-PCR should be performed with an appropriated number of cycles to not saturate.

**Have all data underlying the figures and results presented in the manuscript been provided?**

Reviewer #1: **No: ** This was not provided

Reviewer #2: Yes

Reviewer #3: Yes

PLOS authors have the option to publish the peer review history of their article (what does this mean? ). If published, this will include your full peer review and any attached files.

**Do you want your identity to be public for this peer review?** For information about this choice, including consent withdrawal, please see our Privacy Policy .

Reviewer #1: No

Reviewer #2: No

Reviewer #3: No

---

## [Decision Letter · Decision Letter 1]

26 Jan 2025

Dear Dr Stainier,

We are pleased to inform you that your manuscript entitled "A recombinase-activated ribozyme to knock down endogenous gene expression in zebrafish." has been editorially accepted for publication in PLOS Genetics. Congratulations!

Yours sincerely,

Jeffrey J Essner

Guest Editor

PLOS Genetics

Aleksandra Trifunovic

Section Editor

PLOS Genetics

Aimée Dudley

Editor-in-Chief

PLOS Genetics

Anne Goriely

Editor-in-Chief

PLOS Genetics

Comments from the reviewers (if applicable):

Reviewer's Responses to Questions

**Comments to the Authors:**

Reviewer #1: The authors have nicely addressed my concerns.

Reviewer #2: My comments are adequately addressed and I have no further comments.

Reviewer #3: The revised manuscript is much clearer, and the demonstration of a ribozyme as a KD tool will be a very useful tool for the Zebrafish community and possibly for other vertebrate models.

Although I believe demonstrating the utility of the ribozyme outside the sole albino gene would have been a nice for a manuscript in this particular journal, this is well balanced by a clear characterization of the ribozyme and associated genome engineering at a transgene insertion and at the albino genes.

I am therefore positive for the manuscript quality and usefulness.

I have some minor comments that could enhance the manuscript for PLOS-Genetics.

-The authors argue that other genome editing approaches are variable and depend of the genomic context. The manuscript gives the feeling that the authors solve this problem with the ribozyme, but they do not per se demonstrating this. They rather provide an efficient tool that can be a solution to some of the drawbacks associated with other approaches. This should be clear.

-Similar to above, the authors argue in the introduction that CRISPR is sometime challenging and variable in efficiency. But the ribozyme is not a solution to this problem since it still requires to KI it by CRISPR in an endogenous locus. This is a bit confusing and should be noted in the manuscript, to avoid over-claiming the usefulness of the tool, which have already a great potential. One nice application of the RiboFlip is that it could be included in future transgenes, in order to have the benefit of the transgene expression coupled with potential modulation of its expression, but this not obvious in the manuscript.

-Line 164: the sentence is a bit confusing. Maybe break it in two to make clear that the authors were thinking that several ribozymes would improve efficiency, BUT in fact NOT.

-I am not sure how mosaics are Flipase / Cre / Heatshock experiments, and this should be clear in regards of the UAS reporter system, in order to estimate how many cells expressing the gene of interest, are KD. This is maybe obvious for Zebrafish scientists, but not for other vertebrate researchers.

**Have all data underlying the figures and results presented in the manuscript been provided?**

Reviewer #1: Yes

Reviewer #2: Yes

Reviewer #3: Yes

PLOS authors have the option to publish the peer review history of their article (what does this mean? ). If published, this will include your full peer review and any attached files.

**Do you want your identity to be public for this peer review?** For information about this choice, including consent withdrawal, please see our Privacy Policy .

Reviewer #1: No

Reviewer #2: No

Reviewer #3: No

**Data Deposition**

http://datadryad.org/submit?journalID=pgenetics&manu=PGENETICS-D-24-00919R1

**Press Queries**

---

## [Editor Report · Acceptance letter]

PGENETICS-D-24-00919R1

A recombinase-activated ribozyme to knock down endogenous gene expression in zebrafish.

Dear Dr Stainier,

We are pleased to inform you that your manuscript entitled "A recombinase-activated ribozyme to knock down endogenous gene expression in zebrafish." has been formally accepted for publication in PLOS Genetics! Your manuscript is now with our production department and you will be notified of the publication date in due course.

With kind regards,

Zsofia Freund

PLOS Genetics

On behalf of:
